# Pathogenic spectrum of blood stream infections and resistance pattern in Gram-negative bacteria from Aljouf region of Saudi Arabia

**Altaf Bandy** [1]*, **Abdulrahman Hamdan Almaeen** [2]

**1** Department of Family and Community Medicine, College of Medicine, Jouf University, Sakaka, Kingdom of Saudi Arabia, **2** Department of Pathology, College of Medicine, Jouf University, Sakaka, Kingdom of Saudi Arabia

☯ These authors contributed equally to this work.
* ahbandy@ju.edu.sa, drbanday@gmail.com

**Data Availability Statement:** The data set is published by dryad website https://datadryad.org/stash/dataset/doi:10.5061/dryad.nvx0k6dp9.

## Abstract

### Background

The pathogenic spectrum of bloodstream infections (BSIs) varies across regions. Monitoring the pathogenic profile and antimicrobial resistance is a prerequisite for effective therapy, infection control and for strategies aimed to counter antimicrobial resistance. The pathogenic spectrum of BSIs in blood cultures was analysed, focusing on the resistance patterns of *Acinetobacter baumannii*, *Escherichia coli*, and *Klebsiella pneumoniae*, in Aljouf region.

### Methods

This descriptive cross-sectional study analysed the culture reports of all non-duplicate blood samples collected from January 1 to December 31, 2019. Antibiograms of *A. baumannii*, *E. coli*, and *K. pneumoniae* were analysed for antibiotic resistance. The frequency and percentages of multi-drug, extensively-drug, pan-drug and carbapenem resistance were calculated.

### Results

Of the 222 bloodstream infections, 62.2% and 36.4% were caused by gram-negative and gram-positive bacteria, respectively. Most BSIs occurred in patients aged ≥60 years (59.5%). Among the 103 isolates of the studied Gram-negative bacteria (GNB), 47.6%, 38.8%, and 2.9% were multi-drug, extensively drug and pan-drug resistant respectively. 46% of *K. pneumoniae* isolates were carbapenemase producers. Resistance to gentamycin, 1st–4th generation cephalosporins, and carbapenems was observed for *A. baumannii*. More than 70% of *E. coli* isolates were resistant to 3rd- and 4th-generation cephalosporins. *Klebsiella pneumoniae* presented a resistance rate of >60% to imipenems.

**Funding:** The author(s) received no specific funding for this work.

**Competing interests:** The authors have declared that no competing interests exist.

## Conclusions

Gram-negative bacteria dominate BSIs, with carbapenem-resistant *K. pneumoniae* most frequently detected in this region. Resistant GNB infections make it challenging to treat geriatric patients. Regional variations in antimicrobial resistance should be continually monitored.

## Introduction

Bloodstream infections (BSIs) with resistant microorganisms are associated with a higher risk of mortality of hospitalized patients [1]. Worldwide, the rate of infection with BSIs is increasing, and BSIs are among the top seven causes of mortality in Europe and the 11th leading cause of death in the USA [2]. The risk of BSI increases with age [3], and thus these infections will become more prevalent as the geriatric population increases worldwide. The length of hospital stay is increased in patients with multidrug-resistant (MDR) infections, resulting in an increased risk of mortality and other infections, resulting in high health expenditures [4]. Previous reports have suggested that in the USA or Europe alone, one patient dies every 10 min because of resistant bacterial infections [5]. The pathogenic spectrum and pattern of antimicrobial resistance of BSIs differ across the affected regions owing to the differences in epidemiological and geographic features across regions [6, 7]. *Escherichia coli* and *Staphylococcus aureus* are the most common BSI pathogens according to Korean surveillance data from 2016–2017 [8]. Similar findings were reported by the European Antimicrobial Resistance Surveillance Network data from 2002 to 2009 [9]. However, non-typhoidal salmonella, *Salmonella typhi*, and *Streptococcus pneumoniae* are frequent causes of BSIs in Malawi [10]. Infections with resistant gram-negative bacteria are considered as a severe threat to patients' health worldwide. The resistance of Enterobacteriaceae to 3rd-generation cephalosporins and carbapenems makes these organisms a critical priority requiring urgent attention [11]. The clinical outcome of patients having BSIs caused by carbapenem resistant *Escherichia coli* and *Klebsiella pneumoniae* is poor with a mortality as high as 50%. [12]. A recent study from the Aseer region reported an MDR rate as high as 69% for *Acinetobacter baumannii*, with only 0.05% and 0.04% of these bacteria susceptible to imipenem and meropenem, respectively [13].

Antimicrobial administration and misuse has been linked to the emergence of MDR microorganisms in Saudi Arabia. In 2010, a hospital-based study in Saudi Arabia revealed antimicrobial overuse in intensive care units ranging from 33.2 defined daily doses (DDD) per 100 bed-days for meropenem to 16.0 DDD for piperacillin-tazobactam, compared to 3.75 DDD/100 bed-days for carbapenems and 7.08 for antipseudomonal penicillins [12]. The other reported driver of antimicrobial misuse is over-the-counter sales of antimicrobials by community pharmacies in Saudi Arabia [12].

Information about different types of microorganisms and their resistance will guide physicians, infection control activity, and policy makers in various countries and regions in making evidence-based decisions to overcome antimicrobial resistance [14, 15]. The lack of published literature on BSIs in Saudi Arabia was highlighted in a recent review [16]. Most recent articles of BSIs in Saudi Arabia focused on device- and central line-associated bloodstream infections [17, 18].

Based on this information and the lack of published literature on BSIs from this region, the current, cross-sectional study analysed the antibiograms for the year 2019 from a referral hospital to determine the overall pathogenic spectrum of BSIs, focusing on the antimicrobial

resistance of *A. baumannii*, *E. coli*, and *K. pneumoniae*. This information can guide antimicrobial stewardship programs and infection control activities in hospitals.

## Materials and methods

The Aljouf region is in the north of Saudi Arabia and comprises the three governorates of Sakaka, Qurayyat, and Dumat Al-Jandal. In the Qurayyat and Dumat Al-Jandal governorates, there are 260- and 130-bed general hospitals. The Sakaka governorate is the capital city of the Aljouf region with two specialist hospitals of 300 beds each. Both of these hospitals serve as referral hospitals for the Aljouf region. The total population of this region is 520,737 with 386,663 Saudi residents and 134,074 non-Saudi residents according to 2018 statistics.

The study hospital has a dedicated microbiology unit equipped with an automated Vitek 2 system (bioMérieux, Marcy-lÉtoile, France), BD Phoenix system (BD Biosciences, Franklin Lakes, NJ, USA), MicroScan plus (Beckman Coulter, Brea, CA, USA), and BD BACTEC system (BD Biosciences) for the identification and antimicrobial sensitivity analysis of microorganisms. The BD Phoenix identification system was used for bacterial identification and susceptibility testing. This system is advantageous because it combines the steps of identification, antimicrobial testing, and florescence control. The antimicrobial susceptibility testing was performed and interpreted in accordance with the Clinical and Laboratory Standard Institute recommendations [19].

In this cross-sectional study, data from all non-duplicate blood samples that depicted the culture and sensitivity of *A. baumannii*, *E. coli*, and *K. pneumoniae* from January 1 to December 31, 2019 were analysed. This study included data from hospitalized patients only. Microorganisms were classified as multi drug-resistant (MDR), extensive drug-resistant (XDR), and pan drug-resistant (PDR) as defined in the guidelines of the European Centre for Disease Control [20]. To simplify the results, bacterial isolates showing intermediate-resistant isolates were classified as resistant strains. The phenotypic characterization of carbapenem, potential carbapenem, ESBL and AmpC β-lactamase-producers as provided by Phoenix system was recorded. Carbapenem and potential carbapenem producer were categorised as carbapenem producers. The demographic data on age and gender, hospitalization data like admitting unit and the date sample was received in the laboratory was extracted from the records.

## Consent and research ethics

The research protocol was approved by the Local Committee of Bio Ethics at Jouf University (vide no: 03/04/41 dated January 6, 2020). In this study the culture and sensitivity reports of *A. baumannii*, *E. coli*, and *K. pneumoniae* of admitted patients were analyzed. As a standard practice at this hospital, a verbal consent is taken by the attending nurse in presence of the patients' relative as witness and the same is filled in the laboratory request form. For patients' admitted to intensive care units, the patients' guardian approve or disapprove any procedure and is recorded in the patient files.

## Statistical analysis

The data were analysed with SPSS version 20.0 for Windows (SPSS, Inc., Chicago, IL, USA). The frequencies and percentages MDR, XDR, PDR, ESBl-, AmpC β-lactamase- and carbapenem producers were calculated. Descriptive analysis of the sample distribution, age, gender, and antimicrobial data was performed, and the results are presented as frequencies and percentages.

## Results

A total of 222 non-duplicate BSI samples from hospitalized patients were tested for culture and sensitivity during the study period, most (77.8%) of which were from male and female intensive care units. BSIs were detected in 134 (60.4%) male and 88 (39.6%) female patients. The largest (43.7%) number of samples was processed in the first quarter (January 1 to March 31, 2019) of the year. Gram-negative microorganisms were the most frequent (62.2%) BSI-causing pathogens. *Klebsiella pneumoniae* was the most frequent (28.4%) gram-negative pathogen; *S. aureus* contributed to 11.3% of gram-positive microorganisms and 1.35% of fungal species causing BSI. More than half (59.5%) of BSIs occurred in patients ≥60 years of age. *Acinetobacter baumannii*, *E. coli*, and *K. pneumoniae* were detected in 46.4% (103/222) of BSIs (Table 1).

Among the 138 gram-negative bacteria identified, 103 (76.6%) isolates were positive for *A. baumannii*, *E. coli*, and *K. pneumoniae*. These organisms contributed to 42.7% of infections in the 1st quarter of the study period. Advanced age and being male were risk factors for BSI. Among the three microorganisms, *K. pneumoniae* was the most common (61.1%) gram-negative pathogen causing BSIs. Multi-drug resistance pattern was most frequent among the GNB causing BSIs (Table 2).

Of the studied gram-negative bacteria identified during the study period, all isolates of *A. baumannii* and 36.5% *K. pneumoniae* isolates showed extended drug-resistance. 78.3% of *E. coli*, and 49.2% of *K. pneumoniae* presented with multi drug-resistance. Phenotypic classification showed 46% *K. pneumoniae* isolates as carbapenem producer and 52.2% of *E. coli* as ESBL producers. (Table 3).

*Acinetobacter baumannii* isolates showed resistance to gentamycin, 1st–4th-generation cephalosporins, and carbapenems. A 70.6% resistance rate to trimethoprim-sulfamethoxazole was observed; however, all isolates were sensitive to colistin. More than 90% of *E. coli* isolates showed resistance to 1st-generation cephalosporins, more than 75% to cefuroxime (2nd), and more than 70% to 3rd- and 4th-generation cephalosporins. However, 87% sensitivity to cefoxitin, a 2nd-generation cephalosporin, was observed. Resistance to fluoroquinolones was detected in more than 60% of isolates. The observed sensitivity of *E. coli* was >90% to carbapenems, 82% to piptazobactam, 95% to nitrofurantoin, and 95.7% to amikacin. In *K. pneumoniae*, the observed resistance rate was >90% to 1st -, >80% to 2nd-, 3rd-, and 4th-generation cephalosporins, respectively except cefoxitin a second generation where the observed resistance was > 60%. A >60% resistance rate was observed to imipenems and >65% to fluoroquinolones; however, a sensitivity rate of 77.4% to amikacin was observed. *Escherichia coli* and *K. pneumoniae* were found to be ESBL-producers. Aztreonam, an antibiotic with proven efficacy against ESBLs, showed a resistance rate of 73.9% for *E. coli* and 84.1% for *K. pneumoniae*. Fourteen out of seventeen (82.4%) isolates of *K. pneumoniae* tested exhibited sensitivity to colistin. A limited number of isolates of *A. baumannii* (3/17 isolates), *E. coli* (11/23 isolates) and *K. pneumoniae* (50 /63 isolates) were tested for tigecycline resistance (Table 4). It should be noted that the percentages shown in this table are based on the number of isolates tested against each antibiotic.

## Discussion

The phenomenon of extended antimicrobial resistance of gram negative bacteria is challenging mankind and this phenomenon is observed in all parts of the world. The seriousness of the problem is highlighted in the World Health organizations 'global priority list of antibiotic-resistant bacteria,' document 2017, wherein gram negative organism figure in the critical priority category [11]. Our ability to treat such infections is limited by the currently available

**Table 1. Pathogenic spectrum of BSIs and distribution of blood samples (n = 222).**

| Category | Number (n) | Percentage (%) |
|---|---|---|
| **Gram-negative bacteria n = 138 (62.1%)** | | |
| *K. pneumoniae* | 63 | 28.4 |
| *E. coli* | 23 | 10.4 |
| *A. baumannii* | 17 | 7.6 |
| *P. aeruginosa* | 8 | 3.6 |
| *S. marcescens* | 8 | 3.6 |
| *E. aerogenes* | 6 | 2.7 |
| Others | 13 | 5.8 |
| **Gram-positive bacteria n = 81 (34.5%)** | | |
| *S. aureus* | 25 | 11.3 |
| *Streptococcus species* | 24 | 10.8 |
| *E. faecalis* | 10 | 4.5 |
| *S. capitis* | 6 | 2.7 |
| Others | 16 | 7.2 |
| **Fungi n = 3 (1.4%)** | | |
| *Candida glabrata* | 1 | 0.4 |
| *Candida species* | 2 | 0.9 |
| **Quarter** | | |
| Quarter-1 | 97 | 43.7 |
| Quarter-2 | 44 | 19.7 |
| Quarter-3 | 29 | 13.0 |
| Quarter-4 | 52 | 23.3 |
| **Unit** | | |
| Male & female ICU[a] | 169 | 76.1 |
| Male & female MW[b] | 46 | 20.7 |
| Male & female SW[c] | 6 | 2.7 |
| Others | 1 | 0.4 |
| **Gender** | | |
| Males | 134 | 60.4 |
| Females | 88 | 39.6 |
| **Nationality** | | |
| Saudi | 198 | 89.2 |
| Non-Saudi | 24 | 10.8 |
| **Age** | | |
| ≥60 years | 132 | 59.5 |
| 41–59 years | 47 | 21.2 |
| 21–40 years | 29 | 13.1 |
| ≤20 years | 14 | 6.3 |

[a]ICU = Intensive care unit

[b]MW = Medical ward

[c]SW = Surgical ward

antimicrobials. The problem is further compounded by the lack of development of new anti-microbial agents. Information about the pattern of antimicrobial resistance will guide actions of local and regional bodies to counter antimicrobial resistance. BSIs with resistant bacteria is difficult to treat. Elderly age is an important risk for BSIs, the number of BSIs are bound to

**Table 2. Distribution of BSIs caused by *A. baumannii*, *E. coli*, *and K. pneumoniae* and their resistance pattern (n = 103).**

| Category | Number (n) | Percentage (%) |
|---|---|---|
| **Quarter** | | |
| Quarter-1 | 44 | 42.7 |
| Quarter-2 | 23 | 22.3 |
| Quarter-3 | 15 | 14.6 |
| Quarter-4 | 21 | 20.4 |
| **Gender** | | |
| Males | 60 | 58.3 |
| Females | 43 | 41.7 |
| **Nationality** | | |
| Saudi | 92 | 89.3 |
| Non-Saudi | 11 | 10.7 |
| **Unit** | | |
| Male & female ICU[a] | 92 | 89.3 |
| Male & female MW[b] | 9 | 8.7 |
| Male SW[c] | 2 | 1.9 |
| **Age** | | |
| ≥60 years | 64 | 62.1 |
| 41–59 years | 21 | 20.4 |
| 21–40 years | 14 | 13.6 |
| ≤20 years | 4 | 3.9 |
| **Microorganisms** | | |
| *Klebsiella pneumoniae* | 63 | 61.2 |
| *Escherichia coli* | 23 | 22.3 |
| *Acinetobacter baumannii* | 17 | 16.5 |
| **Resistance pattern** | | |
| MDR [d] | 49 | 47.6 |
| XDR[e] | 40 | 38.8 |
| PDR[f] | 3 | 2.9 |
| Carbapenemase producers | 30 | 29.1 |
| ESBLproducers[g] | 24 | 23.3 |
| AmpC β-lactamase producer | 2 | 1.94 |

[a]ICU = Intensive care unit
[b]MW = Medical ward
[c]SW = Surgical ward
[d]MDR = Multi-drug resistance
[e]XDR = Extended drug-resistance
[f]PDR = Pan drug-resistance
[g]ESBL = Extended beta lactamase

increase as the geriatric population of the world continues to grow. Pathogenic profile of blood stream infections and antimicrobial resistance among gram negative bacteria causing BSIs is not available from this region.

The major finding of the study is that the gram-negative microorganisms dominate (62.2%) BSIs, with *K. pneumoniae* accounting for 28.4% of these infections. Gram-positive bacteria accounted for 34.5% of BSIs involving *S. aureus* in 11.3% of all BSIs. These results contradict those of a recent report on resistance trends in BSI from China (surveillance study 1998–

**Table 3. Resistance pattern and phenotypic classification in _A. baumannii_, _E. coli_, and _K. pneumoniae_ (103).**

| Category | Number (n) | Percentage (%) |
|---|---|---|
| _A. baumannii_ **(17)** | | |
| XDR | 17 | 100.0 |
| _E. coli_ **(23)** | | |
| MDR | 18 | 78.3 |
| Carbapenem producer | 1 | 4.3 |
| **Phenotypic classification** | | |
| AmpC β-lactamase producer | 2 | 8.7 |
| ESBL producers[a] | 12 | 52.2 |
| Carbapenemase producers | 1 | 4.3 |
| _K. pneumoniae_ **(63)** | | |
| MDR[b] | 31 | 49.2 |
| XDR[c] | 23 | 36.5 |
| PDR[d] | 3 | 4.8 |
| **Phenotypic classification** | | |
| Carbapenemase producers | 29 | 46.0 |
| ESBL producers | 12 | 19.0 |

[a] ESBL = Extended beta lactamase

[b] MDR = Multi-drug resistance

[c] XDR = Extended drug-resistance

[d] PDR = Pan drug-resistance

2017), which identified _E. coli_ and _S. aureus_ as the most common BSI-causing pathogens [21] and a study conducted in Australia [22]. However, a previous study in Hubei Province of China reported _E. coli_, _S. aureus_, and _K. pneumoniae_ as frequent BSI-causing pathogens. Reports from Malawi in Africa [10] revealed _Salmonella typhi_ and _S. pneumoniae_ as BSI-causing pathogens, whereas _Pseudomonas aeruginosa_ and _Staphylococcus_ species were more common in Iran [23]. Similarly, Japan has shown a varying pathogenic profile of BSIs, with _E. coli_, _S. aureus_, _Streptococcus_ spp., and _Klebsiella_ spp. as the common organisms [24]. These differing reports on BSI-causing pathogens explain the variations in different regions. Our results agree with a study performed in Saudi Arabia which showed that most general Enterobacteriaceae isolates were _K. pneumoniae_ and _E. coli_ [25].

In general, BSIs showed a rate of 6.3% in patients in the group of ≤ 20 years of age and 59.5% in those ≥60 years of age. Furthermore, males showed a higher risk (60.4%) of BSI compared to females (39.6%). In elderly population the immune system functions less efficiently, so the risk acquiring BSIs increases. In addition, elderly population frequently have comorbidities affecting immunity. This finding agrees with a previous study showing that being elderly and male are risk factors for acquiring BS1 [3, 26]. The number of BSIs is likely to increase in Saudi Arabia given that the population aged >60 years is expected to increase by 18.5% by 2035, [27] making treatment challenging in the absence of new therapeutic options.

Of the 103 gram-negative isolates, 24 were ESBL-producers showing an overall prevalence of 23.3%. Twelve of 23 _E. coli_ isolates were classified as ESBL-producers and 12 of 63 isolates of _K. pneumoniae_ were ESBL-producers. _Klebsiella pneumoniae_ and _E. coli_ have also been reported as major ESBL-producers in hospital settings both within and outside of Saudi Arabia [28–31]. The prevalence of ESBL-producers in the current study is lower than those determined in studies performed in various regions in Saudi Arabia, which showed proportions of 27–70% [28, 32, 33], but higher than that in the Netherlands with a rate of 5% [34]. The main

**Table 4. Antibiotic resistance and susceptibility profiles of *A. baumannii*, *E. coli*, and *K. pneumoniae* (n = 103).**

| Antibiotic | Classification | Name of the microorganism (n = 103) | | | | | |
| | | *A. baumannii* (17) | | *E. coli* (23) | | *K. pneumoniae* (63) | |
| | | n | % | n | % | n | % |
|---|---|---|---|---|---|---|---|
| Amikacin | Resistant | 16 | 94.1 | 1 | 4.2 | 14 | 22.6 |
| | Sensitive | 1 | 5.9 | 22 | 95.7 | 48 | 77.4 |
| Gentamicin | Resistant | 17 | 100.0 | 5 | 21.7 | 25 | 40.3 |
| | Sensitive | 0 | 0 | 18 | 78.3 | 37 | 59.7 |
| Ertapenem | Resistant | 17 | 100.0 | 1 | 4.3 | 39 | 61.9 |
| | Sensitive | 0 | 0 | 22 | 95.7 | 24 | 38.1 |
| Imipenem | Resistant | 17 | 100.0 | 1 | 4.3 | 41 | 65.1 |
| | Sensitive | 0 | 0 | 22 | 95.7 | 22 | 34.9 |
| Meropenem | Resistant | 17 | 100.0 | 0 | 0 | 40 | 63.5 |
| | Sensitive | 0 | 0 | 23 | 100.0 | 23 | 36.5 |
| Cephalothin | Resistant | 17 | 100.0 | 21 | 91.3 | 57 | 90.5 |
| | Sensitive | 0 | 0.0 | 2 | 8.7 | 6 | 9.5 |
| Cefuroxime | Resistant | 17 | 100.0 | 18 | 78.3 | 55 | 87.3 |
| | Sensitive | 0 | 0.0 | 5 | 13.0 | 8 | 12.7 |
| Cefoxitin | Resistant | 17 | 100 | 3 | 13.0 | 40 | 63.5 |
| | Sensitive | 0 | 0.0 | 20 | 87.0 | 23 | 36.5 |
| Ceftazidime | Resistant | 17 | 100.0 | 17 | 73.9 | 53 | 84.1 |
| | Sensitive | 0 | 0.0 | 6 | 26.1 | 10 | 15.9 |
| Ceftriaxone | Resistant | 17 | 100.0 | 17 | 73.9 | 54 | 85.3 |
| | Sensitive | 0 | 0.0 | 6 | 26.1 | 9 | 14.3 |
| Cefepime | Resistant | 17 | 100.0 | 17 | 73.9 | 50 | 80.6 |
| | Sensitive | 0 | 0.0 | 6 | 26.1 | 12 | 19.4 |
| Aztreonam | Resistant | 17 | 100.0 | 17 | 73.9 | 53 | 84.1 |
| | Sensitive | 0 | 0.0 | 6 | 26.1 | 10 | 15.9 |
| Ampicillin | Resistant | 17 | 100.0 | 22 | 95.7 | 63 | 100.0 |
| | Sensitive | 0 | 0.0 | 1 | 4.3 | 0 | 0.0 |
| Amoxicillin and Clavulanate potassium | Resistant | 17 | 100.0 | 17 | 81.0 | 53 | 86.9 |
| | Sensitive | 0 | 0.0 | 4 | 19.0 | 8 | 13.1 |
| Piptazobactam | Resistant | 17 | 100.0 | 4 | 17.4 | 43 | 68.3 |
| | Sensitive | 0 | 0.0 | 19 | 82.6 | 20 | 31.7 |
| Colistin | Resistant | 0 | 0.0 | 0 | 0.0 | 3 | 17.6 |
| | Sensitive | 17 | 100.0 | 4 | 100.0 | 14 | 82.4 |
| Trimethoprim-Sulfamethoxazole | Resistant | 12 | 70.6 | 16 | 69.6 | 43 | 69.4 |
| | Sensitive | 5 | 29.4 | 7 | 30.4 | 19 | 30.6 |
| Nitrofurantoin | Resistant | 17 | 100.0 | 1 | 4.5 | 50 | 80.6 |
| | Sensitive | 0 | 0.0 | 21 | 95.5 | 12 | 19.4 |
| Ciprofloxacin | Resistant | 17 | 100.0 | 15 | 65.2 | 43 | 69.4 |
| | Sensitive | 0 | 0.0 | 8 | 34.8 | 19 | 30.6 |
| Levofloxacin | Resistant | 17 | 100.0 | 14 | 60.9 | 41 | 66.1 |
| | Sensitive | 0 | 0 | 9 | 39.1 | 21 | 33.9 |
| Tigecycline | Resistant | 3 | 100.0 | 0 | 0 | 16 | 32.0 |
| | Sensitive | 0 | 0 | 11 | 100.0 | 34 | 78.0 |

reason for low prevalence in this Dutch study was in the sample characteristics. All the patients in this study were from day care wards and patients on dialysis with an inpatient stay of few days. In our study 76% samples were from intensive care units that might explain the higher prevalence observed in our study. Additionally, two isolates (1.94%) of AmpC β-lactamase-producers were found among gram-negative BSIs. This number is lower than that determined in a recent study conducted in Bisha province in Saudi Arabia which showed rates of above 30% [32] but similar to an earlier study in Saudi Arabia showing a rate of 1% [35]. These organisms should be under constant surveillance, as they have the potential to cause outbreaks in hospital settings.

Antibiogram analysis revealed that more than 90% of E. *coli* isolates showed resistance to 1st generation, more than 75% to cefuroxime and 70% to 2nd–4th generation of cephalosporins. Resistance to fluoroquinolones remains high. The observed sensitivity of E. *coli* to carbapenems was >90%, 82% to piptazobactam, 95% to nitrofurantoin, and 95.7% to amikacin. Similar findings have been observed across Saudi Arabia [32, 36] and China [21]. Study also observed 87% sensitivity to cefoxitin a 2nd-generation cephalosporin, consistent with other reports [21, 36] that have showed a susceptibility rate of >80%.

Carbapenem-resistant (CR) *K. pneumoniae* is consistently reported in Saudi Arabia and neighbouring Gulf Cooperation Council countries [37–40]. Early identification is critical for limiting the spread of CR *K. pneumoniae* in hospital settings [41]. Overall, 29.1% (30/103) of the studied gram-negative bacteria were CR, with 29 isolates of *K. pneumoniae* and one isolate of E. *coli* showing CR. All 30 isolates were recovered from the intensive care unit (ICU), and 27 cases were in the age group of ≥41 years. These findings are consistent with those of previous studies demonstrating that an advanced age and ICU admission are risk factors for infection with CR *K. pneumoniae* [37, 40]. A 20-year surveillance study in China showed that the prevalence of CR *K. pneumoniae* is increasing [21]. The complex environment of the ICU, severe conditions of patients admitted to the ICU, and excessive administration of antimicrobial agents before and during care may have led to the evolution of carbapenem resistance.

Travel facilitates the transmission of resistant Enterobacteriaceae [42]. Although the pilgrimage population is not a factor in the Aljouf region than in the Makkah and Madinah provinces, other factors causing the emergence and dissemination of resistant strains persist. As in other regions of the Kingdom, people from Aljouf routinely travel to Makkah for Umrah throughout the year. Various studies and reports from different regions of the Kingdom have highlighted the spread of resistant strains [37, 43–46]. Furthermore, the expatriate population may import and transmit Enterobacteriaceae with diverse resistance mechanisms [47] Aljouf has an expatriated population of 26% according to 2018 statistical data.

*Klebsiella pneumoniae* isolates showed resistance rates of >90% to 1st-, >80% to 2nd–t0 4th-generation cephalosporins, respectively. A >60% resistance rate to imipenems and >65% rate to fluoroquinolones was observed; however, a 77.4% sensitivity rate was observed to amikacin. Decreased susceptibility to antimicrobials may be a risk factor for increased BSIs [21]. *Klebsiella pneumoniae* presents considerable problems in healthcare settings mainly because of carbapenemase production. Resistant *K. pneumoniae* isolates are being detected with increased frequency in China, USA, and from Saudi Arabia [25, 48]. The finding agrees with the results of a study conducted in Korea [49] which showed increased resistance to beta-lactam and beta-lactamase inhibitors. However, an earlier report in Saudi Arabia analysing antibiotic resistance from 1998 to 2003 revealed low resistance rates for *K. pneumoniae* to aminoglycosides, penicillins, and cephalosporins in 2007 [50]. Thus, *K. pneumoniae* resistance to antimicrobials has greatly increased, creating an important threat in Saudi Arabia. In a recent study, MDR strains co-harbouring the NDM-1, KPC, and OXA-48 genes, which are triple carbapenemase genes, were isolated [25]. We found that the susceptibility to amikacin was >77%,

supporting the results of a previous study in Saudi Arabia [51]; however, this susceptibility is lower than that in China [21].

Currently, few treatment options are available for *A. baumannii* strains that have developed resistance to broad-spectrum antimicrobial agents [52]. This infection is widely disseminated in ICUs because it can colonize a range of surfaces and withstand harsh environments [53]. Forty isolates of XDR were observed, with all 17 isolates of *A. baumannii* and 23 isolates of *K. pneumoniae*. All 17 isolates of *A. baumannii* were recovered from ICU samples. *Acinetobacter baumannii* showed 100% resistance to gentamycin, penicillins, cephalosporins, fluoroquinolones, and carbapenems. Colistin was 100% effective and trimethoprim-sulfamethoxazole was 29.4% effective against *A. baumannii* isolates. Increased antimicrobial resistance of *A. baumannii* has been reported throughout Saudi Arabia [54]. The only available report on *A. baumannii* resistance from the Aljouf region observed a 7.1% resistance rate to commonly prescribed antimicrobial agents. The 100% resistance to carbapenems observed in this study agrees with the results of Al-Obeid obtained in 2015 which showed a drastic change in resistance to meropenem and imipenem from 19% and 36% in 2006 to 89% and 91.7% in 2012, respectively [55]. A recent study by Al-Agamy et al. in 2018 revealed a similar phenomenon in carbapenem resistance in *A. baumannii* [13]. The current study found a high resistance rate of *A. baumannii* compared to an earlier report from the Aljouf region, [56] which may be attributed to changes in the mechanisms of resistance over time and excessive antimicrobial misuse. In 2010, a hospital-based study in Saudi Arabia revealed antimicrobial overuse in intensive care units ranging from 33.2 DDD per 100 bed-days for meropenem to 16.0 DDD for piperacillin-tazobactam compared to 3.75 DDD/100 bed-days for carbapenems and 7.08 for antipseudomonal penicillins [12]. These findings highlight the need for developing effective inpatient antimicrobial and infection control guidelines to limit the emergence and spread of resistant strains in hospitals.

This is the first study from Aljouf region that adds to the existing literature on the pathogenic profile of BSIs and their resistance patterns, focusing on resistance of *A. baumannii*, *E. coli*, and *K. pneumoniae*. This study has highlighted that BSIs are dominated by gram negative bacteria and will now serve as a base line for future evaluation of antimicrobial resistance in gram negative organisms causing BSIs in the region. The important limitation of the study is that it analysed antibiograms of BSI caused by gram negative bacteria from a single centre. The other limitation is lack of molecular characterization of this observed resistance. The results suggest that future research on BSI causing pathogens should be expanded to cover other regions in order to get a comprehensive understanding on pathogenic profile and antimicrobial resistance pattern across regions with molecular characterization of resistance. The development of resistant strains presents huge challenges to treating physicians. Adherence to guidelines for initiating empirical, antimicrobial therapy in hospital setting is of utmost importance. Furthermore, research should also focus on assessing the advantages of establishing a rapid, culture-free bacterial identification and sensitivity analysis that will not only help early initiation of appropriate antimicrobial therapy for better patient outcome but will also defer bacterial resistance. The other area of improvement is effective antimicrobial resistance surveillance especially from intensive care units and timely dissemination of information for limiting the spread of infection.

## Conclusions

Gram-negative organisms have emerged as a leading cause of BSIs in the region, as most infections were caused by CR *K. pneumoniae* and XDR *A. baumannii*. This high prevalence of resistant BSI-causing pathogens is a serious challenge to infection control in hospital settings and

causes major issues in treatment, particularly in elderly patients. Resistance to carbapenems highlights the issue of non-compliance with antimicrobial drug prescriptions in hospital settings. Dispensing of antimicrobials should be strictly regulated, and over-the-counter sales of antimicrobials should be discouraged. Furthermore, the increased trend in global dissemination and regional variations in resistance emphasize the need for intensified surveillance of CR *K. pneumoniae* in this region.

## Acknowledgments

The author thanks Tarek El-Metwally Dabah for his continuous encouragement for this research. The author would also like to thank Editage for English language editing.

## Author Contributions

**Conceptualization:** Altaf Bandy.

**Data curation:** Altaf Bandy.

**Formal analysis:** Altaf Bandy.

**Methodology:** Altaf Bandy.

**Resources:** Abdulrahman Hamdan Almaeen.

**Writing – original draft:** Abdulrahman Hamdan Almaeen.

**Writing – review & editing:** Abdulrahman Hamdan Almaeen.

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
