## [Decision Letter · Decision Letter 0]

16 Apr 2020

PONE-D-20-05280

Pathogenic Spectrum of Blood Stream Infections and Resistance Pattern in Gram-Negative Bacteria

PLOS ONE

Dear Dr Bandy,

Thank you for submitting your manuscript to PLOS ONE. After careful consideration, we feel that it has merit but does not fully meet PLOS ONE’s publication criteria as it currently stands. Therefore, we invite you to submit a revised version of the manuscript that addresses the points raised during the review process.

We would appreciate receiving your revised manuscript by May 31 2020 11:59PM. To enhance the reproducibility of your results, we recommend that if applicable you deposit your laboratory protocols in protocols.io, where a protocol can be assigned its own identifier (DOI) such that it can be cited independently in the future. For instructions see: http://journals.plos.org/plosone/s/submission-guidelines#loc-laboratory-protocols

We look forward to receiving your revised manuscript.

Kind regards,

Grzegorz Woźniakowski, PhD ScD

Academic Editor

PLOS ONE

2. Please provide additional details regarding participant consent. In the ethics statement in the Methods and online submission information, please ensure that you have specified (1) whether consent was informed and (2) how verbal consent was documented and witnessed.

Reviewers' comments:

Reviewer's Responses to Questions

**Comments to the Author**

1. Is the manuscript technically sound, and do the data support the conclusions?

Reviewer #1: Yes

Reviewer #2: Yes

2. Has the statistical analysis been performed appropriately and rigorously? 

Reviewer #1: Yes

Reviewer #2: No

3. Have the authors made all data underlying the findings in their manuscript fully available?

Reviewer #1: Yes

Reviewer #2: Yes

4. Is the manuscript presented in an intelligible fashion and written in standard English?

Reviewer #1: Yes

Reviewer #2: Yes

5. Review Comments to the Author

Reviewer #1: Pathogenic Spectrum of Blood Stream Infections and Resistance Pattern in Gram-Negative Bacteria

The study conducted by authors presents data about blood strain infection bacteria profiles and their antibiotic resistance. Analysis of samples (n=222) collected from hospitals in Saudi Arabia (Aljouf region) revealed valuable information of types of bacteria (advantage of Gram negative) their genus (K. pneumoniae, E. coli, A. baumannii) and their antbiotic response profile.

Authors indicated, that bacteria originated from every geographic region may differ in antibiotic resistance. In light of this, work done by authors presents a novelty.

Paper is well organised and presents reliable data supported by other studies. However I suggest several minor remarks:

1. If it is possible please specify the region (country) in the title of manuscript

2.

a) lines 68-70: The pathogenic spectrum and pattern of antimicrobial resistance of BSIs

differ across the affected regions owing to the differences in epidemiological and geographic features across regions

b) line 77: In Saudi Arabia, 29% of Escherichia coli strains produce […]

c) line 78-79: In the sentence „For Klebsiella pneumoniae, this rate is 65%, and the reported mortality caused by these organisms remains high” please indicate percentage of mortality rate

d) Table 2, sections Age and Microorganisms: please sort data from the highest to lowest as in the other sections

e) Table 4: Whenever it’s possible I suggest to avoid commercial names of antibiotics such (i.e. Augmentin) and focus on active substances: amoxicillin and clavulanate potassium etc.

f) Table 4, section Tygecycline: According to data shown in the table there was 3 resistant cases which is 17,6% of all cases, and there was no sensitive strains at all (0 –in the table). Where is 82,4% cases? Could you explain that?

I recommend to publish this paper after applying indicated remarks.

Reviewer #2: The manuscript presents interesting data on Pathogenic Spectrum of Blood Stream Infections and Resistance Pattern in Gram-Negative Bacteria. However, i have the following observations.

1. The use of abbreviations in the abstract should be corrected.

2. The data can be explored further by applying non-parametric statistics such as Chi square especially in Table 2 (where applicable). That way the difference between the groups can be better appreciated instead of the percentages. In such cases, provide the p-values deduced as well.

3. There are a lot of result repetitions in the discussion. Also, the discussions are un-conclusive; stating that the values are higher or lower than previous studies alone does not suffice, reasons should be given e.g is it due to variations in sample size, evaluation methods as so on. The entire discussion should be re-written based on this.

6. PLOS authors have the option to publish the peer review history of their article (what does this mean?). If published, this will include your full peer review and any attached files.

Reviewer #1: No

Reviewer #2: No

---

## [Author Response · Author response to Decision Letter 0]

27 Apr 2020

Response to editor comments 

1. The anonymized data supporting the findings has been deposited in ‘Dyrad’ repository. 

For private access during this review period, please visit this temporary link: https://datadryad.org/stash/share/vckG887rWbsoIYu-Elabtk-z_BTIm4WvwThSQnGD6hI.

The dataset has been assigned a unique identifier, called a DOI (doi:10.5061/dryad.nvx0k6dp9) and will be available after acceptance. 

2. The authors have modified the manuscript to meet PLOS ONE’s style requirements.

3. The details regarding ethical approval and consent have been incorporated in the methods section of the revised manuscript.

4. Point by point response to reviewer queries is submitted in the relevant section of submission path.

Response comments Reviewer comments (Reviewer-1)

1. If it is possible please specify the region (country) in the title of manuscript

Response: The region and country has been specified in the title of manuscript.

2. line 78-79: In the sentence „For Klebsiella pneumoniae, this rate is 65%, and the reported mortality caused by these organisms remains high” please indicate percentage of mortality rate.

Response: The statement has been modified and now includes the percentage of mortality rate.

3. Table 2, sections Age and Microorganisms: please sort data from the highest to lowest as in the other sections.

Response: In the section the suggested changes have been made.

4. Table 4: Whenever it’s possible I suggest to avoid commercial names of antibiotics such (i.e. Augmentin) and focus on active substances: amoxicillin and clavulanate potassium etc

Response: The commercial name of antibiotic Augmentin is changed to the active substance.

5. Table 4, section Tygecycline: According to data shown in the table there was 3 resistant cases which is 17,6% of all cases, and there was no sensitive strains at all (0 –in the table). Where is 82,4% cases? Could you explain that?

Response: I thank the reviewer for this important query. This comment was immensely beneficial. The correction has been made and a statement included in the results section.

Response to reviewer-2

1. The use of abbreviations in the abstract should be corrected.

Response: All abbreviations have been removed from abstract

2. The data can be explored further by applying non-parametric statistics such as Chi square especially in Table 2 (where applicable). That way the difference between the groups can be better appreciated instead of the percentages. In such cases, provide the p-values deduced as well.

Response: Thanks for these comments. We wish to keep this table as it is because applying chi-square test does not have any impact on the understanding of the findings. Besides we wish to keep it as simple descriptive as possible.

3. There are a lot of result repetitions in the discussion. Also, the discussions are un-conclusive; stating that the values are higher or lower than previous studies alone does not suffice, reasons should be given e.g is it due to variations in sample size, evaluation methods as so on. The entire discussion should be re-written based on this.

Response: In its present form, every section of the discussion presents the major findings of the paper followed by comparison and reasoning in case of any observed variation in findings.

Based on the suggestion the repetition of the results in this section have been reduced and more focus has been paid to reasoning

---

## [Decision Letter · Decision Letter 1]

12 May 2020

Pathogenic spectrum of blood stream infections and resistance pattern in Gram-negative bacteria from Aljouf region of Saudi Arabia

PONE-D-20-05280R1

Dear Dr. Bandy,

We are pleased to inform you that your manuscript has been judged scientifically suitable for publication and will be formally accepted for publication once it complies with all outstanding technical requirements.

With kind regards,

Grzegorz Woźniakowski, PhD ScD

Academic Editor

PLOS ONE

Additional Editor Comments (optional):

Reviewers' comments:

Reviewer's Responses to Questions

**Comments to the Author**

1. If the authors have adequately addressed your comments raised in a previous round of review and you feel that this manuscript is now acceptable for publication, you may indicate that here to bypass the “Comments to the Author” section, enter your conflict of interest statement in the “Confidential to Editor” section, and submit your "Accept" recommendation.

Reviewer #1: All comments have been addressed

Reviewer #2: All comments have been addressed

2. Is the manuscript technically sound, and do the data support the conclusions?

Reviewer #1: Yes

Reviewer #2: Yes

3. Has the statistical analysis been performed appropriately and rigorously? 

Reviewer #1: Yes

Reviewer #2: N/A

4. Have the authors made all data underlying the findings in their manuscript fully available?

Reviewer #1: Yes

Reviewer #2: Yes

5. Is the manuscript presented in an intelligible fashion and written in standard English?

Reviewer #1: Yes

Reviewer #2: Yes

6. Review Comments to the Author

Reviewer #1: Dear Author

Thank you for applying indicated remarks.

I've got no further comments.

I recommend to publish manuscript.

Kind regards.

Reviewer #2: (No Response)

7. PLOS authors have the option to publish the peer review history of their article (what does this mean?). If published, this will include your full peer review and any attached files.

Reviewer #1: No

Reviewer #2: No

---

## [Editor Report · Acceptance letter]

22 May 2020

PONE-D-20-05280R1 

Pathogenic spectrum of blood stream infections and resistance pattern in Gram-negative bacteria from Aljouf region of Saudi Arabia 

Dear Dr. Bandy:

I am pleased to inform you that your manuscript has been deemed suitable for publication in PLOS ONE. Congratulations! Your manuscript is now with our production department. 

With kind regards,

on behalf of

Prof. Grzegorz Woźniakowski 

Academic Editor

PLOS ONE